# Oxoglutarate Carrier Inhibition Reduced Melanoma Growth and Invasion by Reducing ATP Production

**DOI:** 10.3390/pharmaceutics12111128

**Published:** 2020-11-23

**Authors:** Jae-Seon Lee, Jiwon Choi, Seon-Hyeong Lee, Joon Hee Kang, Ji Sun Ha, Hee Yeon Kim, Hyonchol Jang, Jong In Yook, Soo-Youl Kim

**Affiliations:** 1Division of Cancer Biology, Research Institute, National Cancer Center, Goyang, Gyeonggi-do 10408, Korea; ljs891109@gmail.com (J.-S.L.); shlee1987@gmail.com (S.-H.L.); wnsl2820@gmail.com (J.H.K.); jsha9595@gmail.com (J.S.H.); 74790@ncc.re.kr (H.Y.K.); hjang@ncc.re.kr (H.J.); 2Department of Oral Pathology, Oral Cancer Research Institute, Yonsei University College of Dentistry, Seoul 03722, Korea; edccjw3235@yuhs.ac (J.C.); jiyook@yuhs.ac (J.I.Y.)

**Keywords:** oxoglutarate carrier, malate-aspartate shuttle, cancer metabolism, ATP production

## Abstract

Recent findings indicate that (a) mitochondria in proliferating cancer cells are functional, (b) cancer cells use more oxygen than normal cells for oxidative phosphorylation, and (c) cancer cells critically rely on cytosolic NADH transported into mitochondria via the malate-aspartate shuttle (MAS) for ATP production. In a spontaneous lung cancer model, tumor growth was reduced by 50% in heterozygous oxoglutarate carrier (OGC) knock-out mice compared with wild-type counterparts. To determine the mechanism through which OGC promotes tumor growth, the effects of the OGC inhibitor *N*-phenylmaleimide (NPM) on mitochondrial activity, oxygen consumption, and ATP production were evaluated in melanoma cell lines. NPM suppressed oxygen consumption and decreased ATP production in melanoma cells in a dose-dependent manner. NPM also reduced the proliferation of melanoma cells. To test the effects of NPM on tumor growth and metastasis in vivo, NPM was administered in a human melanoma xenograft model. NPM reduced tumor growth by approximately 50% and reduced melanoma invasion by 70% at a dose of 20 mg/kg. Therefore, blocking OGC activity may be a useful approach for cancer therapy.

## 1. Introduction

Recently, we reported that up to 80% of the total ATP production in melanoma and lung cancer cells [1], and about 40% of the ATP production in pancreatic cancer cells [2,3], depends on cytosolic NADH and the malate-aspartate shuttle (MAS). The MAS transfers cytosolic NADH into mitochondria for ATP production through oxidative phosphorylation (OxPhos) in the mitochondrial membrane [1]. The MAS consists of four metabolic enzymes, glutamic-oxaloacetic transaminase (GOT) 1 and 2 and malate dehydrogenase (MDH) 1 and 2, and two antiporters, oxoglutarate carrier (OGC, oxoglutarate/malate antiporter, *SLC25A11*) and mitochondrial aspartate-glutamate carrier (AGC1) (Figure 1A) [1,4]. Knock-down of OGC reduced ATP production by 80% and inhibited the growth of lung and melanoma cancer cells by over 90% [1]. ATP depletion by more than 50% induces cell cycle arrest and cell death in a time-dependent manner in cancer cells [5,6,7,8]. Furthermore, in in vivo experiments, heterozygous OGC knock-out mice showed 50% less spontaneous tumor development in the *KRAS^LA2^* lung cancer model [1]. Blocking OGC may selectively inhibit cancer growth by reducing ATP production in cancer cells because cancer cells rely on the MAS for ATP generation while normal cells do not [1,2]. 

A specific mitochondrial transport system for 2-oxoglutarate was first proposed by Dr. Chappell in 1967 [9], and the biochemical properties of this system, including its structure and activity, were recently reviewed by Dr. Fiermonte [10]. OGC has a similar structure to that of mitochondrial ADP/ATP carriers, although a three-dimensional (3D) study of the OGC structure by X-ray crystallography failed [10]. Inhibitors of OGC were therefore developed based on the starting compounds of the biochemical reaction instead of the OGC structure. Most OGC inhibitors are derived from substrate analogues such as succinate, butylmalonate, and phthalonate [11,12], which do not bind to the translocation site but do block transport. OGC contains three cysteine residues at positions 184, 221, and 224 that can create S-S bridges with sulfhydryl reagents [13]. Dr. Palmieri’s group found that both mercurials and maleimides integrated specifically with Cys184 [13]. This binding was associated with inhibition of the OGC active conformation. The degree of OGC inhibition by *N*-phenylmaleimide (NPM) binding to OGC was enhanced in the presence of OGC substrates [13]. This suggests that a substrate-induced conformational change in OGC increases the reactivity of Cys184 to sulfhydryl reagents such as NPM [13]. NPM inhibited OGC transport activity with a 50% inhibitory concentration (IC_50_) of 1.25 mmol/min/g [13], and NPM analogues inhibited the proliferation of H460 cancer cells with IC_50_ values of 0.84–9 μM using in vitro assay system using reconstituted liposome with purified OGC [14].

In this study, we investigated whether OGC inhibition with NPM inhibited cancer growth by reducing ATP production. Although NPM itself is known to have some off-target effects, this study demonstrates the potential therapeutic efficacy of OGC inhibitors.

## 2. Materials and Methods

### 2.1. Cell Culture

Tumor spheres (TS) generated from the UACC-62 and B16F10 melanoma cell lines were used in this study. UACC-62 cells were obtained from the US National Cancer Institute (NCI; Bethesda, MD, USA) (MTA 1-2702-09) and B16F10 cells (CRL-6475) were obtained from the American Type Culture Collection (ATCC; Manassas, VA, USA). For TS culture, cells were cultured in TS complete media, which was composed of Dulbecco’s modified Eagle’s medium/F12 (SH30023.01, Hyclone, Logan, UT, USA), B27 supplement (17504044, Thermo Fisher Scientific, Waltham, MA, USA), 20 ng/mL basic fibroblast growth factor (F0291, Sigma-Aldrich, St. Louis, MO, USA), and 20 ng/mL epidermal growth factor (E9644, Sigma-Aldrich, St. Louis, MO, USA). 

### 2.2. Homology Modeling and Molecular Docking

To obtain a homology model of OGC, we used the nuclear magnetic resonance (NMR)-based structure of mitochondrial uncoupling protein 2 (UCP2; Protein Data Bank (PDB) ID, 2LCK) as a template structure. The sequence alignment and homology modeling procedure were executed using Prime (Schrödinger, New York, NY, USA, 2020) [15]. The overall quality of the modeled structure was assessed using a Ramachandran plot. The active binding sites of OGC were identified using the modeled protein structure in the SiteMap program of the Schrödinger software [16]. To examine the binding interactions of the protein–ligand complexes, molecular docking studies were performed using the Glide software (Schrödinger, New York, NY, USA), which uses an optimized potential for liquid simulations (OPLS)-2005 force field, and refinement was carried out as per the recommendations of the Schrödinger Protein Preparation Wizard. LigPrep (Schrödinger, New York, NY, USA) was used to generate 3D structures of the ligands. The active grid was generated using the Receptor grid application in the Glide module. On a defined receptor grid, flexible docking was performed using the standard precision mode of Glide [17]. The best docking pose for a compound was selected based on the best-scoring conformations from Glide, the binding patterns, and visual inspection.

### 2.3. Sulforhodamine B Assay (SRB): Cell Growth Assay

Cancer cells were counted, and approximately 2 × 10^4^ cells per well were seeded in 96-well cell culture plates (Corning Inc., Corning, NY, USA). After incubation at 37 °C in a humidified atmosphere with 5% CO_2_ for 72 h, cells were treated with the indicated concentrations of NPM. The assay was performed following to the previously established method [2].

### 2.4. Measurement of the NADH/NAD^+^ Ratio and ATP Levels

To quantify NADH/NAD in cell lines, the Ultra-GloTM Recombinant Luciferase assay kit (Promega, #G9071) was used according to the manufacturer’s instructions. Briefly, the cyclic enzyme included in the kit converts NAD^+^ to NADH, which subsequently activates a reductase that converts pro-luciferin to luciferin. The samples were subsequently detected with Ultra-GloTM r-Luciferase. To this end, the cells were seeded into a 96-well culture plate at a density of 10^4^ cells/well and incubated for 24 h, then treated with NPM for 72 h. Subsequently, 50 µL of NAD/NADH-GloTM Detection Reagent and an equal volume of sample were incubated at room temperature for 30 min. To measure ATP levels in cell lines, the cell titer-Glo 2.0 assay (Promega, #G9241) was used according to the manufacturer’s instructions. The cells were seeded into a 96-well culture plate at a density of 10^5^ cells/well and incubated for 24 h and treated with NPM for 72 h. A volume of CellTiter-Glo^®^ 2.0 Reagent equal to the volume of ATP standard present in each well was added. The mixed contents were incubated at room temperature for 10 min and luminescence was measured.

### 2.5. Measurement of Apoptosis

Tumor cells were incubated with or without NPM. The cells were collected, washed with cold PBS, centrifuged at 1400 rpm for 3 min, and resuspended in binding buffer from a kit (556547, BD Biosciences, San Jose, CA, USA) at a density of 1 × 10^6^ cells/mL. Cells (1 × 10^5^ cells in a 100 μL volume) were transferred to a 5 mL culture tube, and 5 μL each of annexin V-FITC and propidium iodide (PI) were added. The cells were gently vortexed and incubated for 15 min at room temperature in the dark. A total of 400 μL of binding buffer was added to each tube, and the samples were analyzed by flow cytometry (FACSCalibur BD Biosciences, San Jose, CA, USA).

### 2.6. Measurement of Mitochondrial Membrane Potential (∆ψm)

Mitochondrial membrane potential (MMP) was assessed by measuring the mean fluorescence intensity of tetramethylrodamine ester (TMRE) loaded cells. TMRE (87917, Sigma-Aldrich, St. Louis, MO, USA) is a fluorescence probe that specifically accumulates within mitochondria in an MMP-dependent manner. Cells were plated in a 100 mm plate and treated as indicted. Twenty minutes prior to the end of each treatment, 100 nM TMRE was added to the culture medium. Cells were washed two times with ice-cold PBS. Cells were collected immediately for flow cytometric analysis (FACSCalibur, BD Biosciences, San Jose, CA, USA) of fluorescence intensity using the 585 nm (FL-2) channel.

### 2.7. Immunohistochemistry

Immunohistochemistry was performed on a Ventana Discovery XT automated staining instrument (Ventana Medical Systems, Tucson, AZ, USA) followed by the established method [2]. Staining was performed with a Ki-67 antibody (ab15580; Abcam, Cambridge, UK) and KI-67 positive cells were quantified using ImageJ software (64-bit Java 1.8.0_112).

### 2.8. XF Cell Mito Stress Analysis

Cells were treated with the indicated drugs for 24 h. For determination of the oxygen consumption rate (OCR), cells were incubated in XF base medium supplemented with 10 mM glucose, 1 mM sodium pyruvate, and 2 mM of L-glutamine, and were equilibrated in a non-CO_2_ incubator for 1 h before starting the assay. The samples were incubated for 3 min and then data were acquired for 3 min using the XFe96 extracellular flux analyzer (Seahorse Bioscience, North Billerica, MA, USA). Oligomycin (0.75 µM), carbonyl cyanide-4-(trifluoromethoxy) phenylhydrazone (FCCP; 1 µM), and rotenone/antimycin A (0.5 µM) (103015-100, Agilent Technologies, Santa Clara, CA, USA) were injected at the indicated time points. Finally, the OCR was normalized to the cell number, as determined by CCK-8 assay.

### 2.9. CCK-8 Assay

Cancer cells were counted, and approximately 5 × 10^3^ cells per well were seeded in 96-well cell culture plates (Corning Inc., Corning, NY, USA). After incubation at 37 °C in a humidified atmosphere with 5% CO_2_ for 72 h, cells were treated with the indicated concentrations of NPM. At the end of XF Cell Mito Stress Analysis, 10 μL of CCK-8 reagent (ALX-850-039, Enzo Life Sciences, Farmingdale, NY, USA) was added to each well, and the optical density (OD) at 450 nm was measured using a multifunction microplate reader (Infinite M200 Pro, Tecan, Männedorf, Switzerland) after incubation for 1 h at 37 °C. 

### 2.10. Invasion Assay

The upper compartments of 8 mm Transwells (6.5 mm diameter; Coastar Corp., Cambridge, MA, USA) were precoated with Matrigel (1 mg/mL). Cells (10^5^ cells) were suspended in DMEM and placed in the upper compartments of the Transwells, and the lower compartments were filled with DMEM supplemented with 3% FBS. After 24 h, the filters were washed with PBS and fixed with methanol. Migrated cells on the filter membrane were stained using a Diff-Quik staining kit (38721, Sysmex, Kobe, Japan). Each assay was conducted at least three times, and three random fields under 20× magnification were analyzed for each filter membrane.

### 2.11. Preclinical Xenograft Model

Balb/c-nu mice (Orient, Seoul, Korea) were between 6 and 8 weeks of age before tumor induction. This study was reviewed and approved by the Institutional Animal Care and Use Committee (IACUC) of the National Cancer Center Research Institute, which is an Association for Assessment and Accreditation of Laboratory Animal Care (AAALAC) International-accredited facility that abides by the Institute of Laboratory Animal Resources guidelines (protocol NCC-19-195). Mice were inoculated with UACC-62 cells (1 × 10^7^) in 100 μL PBS subcutaneously using a 1 mL syringe. After 1 week, the mice were divided into two groups, a control group treated with vehicle and an NPM-treated group (*n* = 5 mice/group). Vehicle (5% dimethyl sulfoxide (DMSO) and 10% Kolliphor in PBS; 100 μL) and NPM (20 mg/kg) were administered orally once per day, 5 days/week, for 3 weeks. The primary tumor size was measured every week using calipers. The tumor volume was calculated using the formula V = (A × B^2^)/2, where V is the volume (mm^3^), A is the long diameter, and B is the short diameter.

### 2.12. Syngeneic Lung Tumor Metastasis Model

This study was reviewed and approved by the IACUC of the National Cancer Center Research Institute (protocol NCC-20-557). To determine the effect of NPM on tumor metastasis in vivo, the formation of lung metastases was assessed in C57BL/6 mice injected intravenously with B16F10 cells (1 × 10^5^) via the tail vein. Three weeks after injection of B16F10 cells, the mice were sacrificed by CO_2_ asphyxiation, images of the lungs were captured, and the lung metastases were counted. Tissue specimens from each group were fixed in formalin and embedded in paraffin for histologic examination. Metastatic lesions were quantified using ImageJ software (64-bit Java 1.8.0_112).

### 2.13. hERG K^+^ Channel Binding Assay

E-4031 (M5060, Sigma-Aldrich, St. Louis, MO, USA) compound was used as a positive control. Membrane containing the hERG channel was mixed with the tracer for 4 h. The fluorescence intensity in the presence of NPT (excitation at 530 nm, emission at 590 nm) was measured by Synergy Neo (Biotek, Winooski, VT, USA) using the hERG Fluorescence Polarization Assay kit (PV5365, Thermo Fisher Scientific) and compared with the fluorescence intensity of the DMSO solvent control. The hERG assay was performed by a licensed contract research organization, Daegu-Gyeongbuk Medical Innovation Foundation (Daegu, Korea).

### 2.14. Statistical Analysis

Statistical analysis was performed using the Student’s *t*-test as appropriate. Tumor growth and tumor weight was analyzed statistically by two-way analysis of variance (ANOVA) tests using GraphPad PRISM 5 (GraphPad Software, San Diego, CA, USA).

## 3. Results

### 3.1. Molecular Docking of NPM to OGC 

Tumor spheres (TSs) were generated from the UACC-62 and B16F10 melanoma cell lines and used in this study to mimic a 3D culture system. The OGC inhibitor NPT was used to test whether OGC inhibition could block the MAS, reduce ATP production, and inhibit cancer growth [13]. Unlike healthy cells, cancer cells utilize the MAS to transfer NADH from the cytosol to mitochondria. The MAS is composed of two antiporters, OGC and AGC1, as well as GOT1 and 2 and MDH1 and 2 (Figure 1A). Although OGC has a similar structure to that of mitochondrial ADP/ATP carriers (PDB ID: 1OKC), there is no crystal structure for the human OGC in the PDB database. By sequence similarity searching, the crystal structure of mouse mitochondrial uncoupling protein 2 (UCP2; PDB ID: 2LCK) was found to be most similar to the sequence of human OGC protein [18]. To obtain homology structure model of human OGC, we used the crystal structure of mouse UCP2 as a template. The Cys184 residue of mitochondrial OGC was found to be more accessible to sulfhydryl reagents when substrate and residues previously defined to be essential for substrate binding were added to the model [13]. Thus, we considered the binding site around Cys184 as the specific target site for the sulfhydryl reagent NPM. Grid-based docking of NPM around Cys184 in the OGC homology model produced a configuration in which the hydroxy group of NPM forms hydrogen bonds with Arg146 of the binding site (Figure 1B). As illustrated, the N-phenyl ring docked into a hydrophobic cavity formed by residues Val141, Ile145, Leu180, and Pro186 (Figure 1C,D). These residues involved in the interaction of OGC with NPM are likely essential for transport function and conformational changes. In particular, Arg146, which is involved in the salt-bridge network, is likely to play a major role in opening and closing the matrix gate to alter the conformational state of the carrier.

### 3.2. The Effect of NPM on Mitochondrial Activity and ATP Production in Cancer Cells

Recently, we showed that knock-down of OGC reduced ATP production by up to 80%, with a concomitant reduction in mitochondrial activity [1]. To test whether NPM reduces ATP production by inhibiting OGC and therefore inhibiting mitochondrial activity, the OCR and ATP production were analyzed using Seahorse analyzer after NPM treatment (Figure 2). NPM treatment for 24 h reduced the OCR by about 20% and 33% in UACC-62 and B16F10 cells, respectively (Figure 2A). ATP production was also reduced by 28% and 24% in UACC-62 and B16F10 cells compared with nontreated control cells (Figure 2A). NPM treatment for 72 h reduced the OCR by approximately 50% and 60% in UACC-62 and B16F10 cells, respectively, which correlated with an approximately 52% and 57% reduction in ATP production compared with control UACC-62 and B16F10 cells (Figure 2A). These results suggest that cancer cells rely on OxPhos, which consumes oxygen to produce ATP using NADH from the cytosol, which is in turn transferred into mitochondria by OGC. To test whether the reduction in OCR by NPM is related to a decrease in OxPhos, mitochondrial membrane potential was measured by tetramethylrhodamine-ethylester (TMRE) staining (Figure 2B). Mitochondrial membrane potential was decreased by NPM treatment, which showed a dose and time dependence between 24 and 48 h at 10 μM (Figure 2B). NPM treatment at 10 μM for 48 h decreased mitochondrial activity to about 10% of the control in UACC-62 and B16F10 cells (Figure 2B). This implies that OGC inhibition reduces NADH flux into mitochondria from the cytosol, which reduces ATP production by OxPhos. To test whether NPM inhibits OGC, we analyzed the OCR and ATP production after treatment with 20 µM NPM for 24 h in OGC knock-down B16F10 cells (Figure 2C). OGC knock-down alone decreased the OCR and the ATP production, but there was no additional effect of NPM in OGC knock-down cells (Figure 2C). Therefore, NPM reduces ATP production by inhibiting OGC in melanoma cells.

### 3.3. The Inhibitory Effect of NPM Treatment on Proliferation of Tumor Spheres

To test whether NPM can regulate cancer growth through OGC inhibition, cell proliferation was measured by Cell Counting Kit-8 (CCK-8) assay, a sensitive colorimetric assay for the determination of cell viability. Treatment with 20 μM NPM decreased TS proliferation to about 70% and 55% of the control in UACC-62 and B16F10 cells, respectively (Figure 3A). NPM is known as an OGC inhibitor, which transfers malate from the cytosol to mitochondria. The transferred malate is converted to oxaloacetate by MDH2, resulting in NADH production. Therefore, we tested whether NPM reduced the NADH and ATP level. NADH and ATP were reduced dose-dependently by NPM treatment for 72 h. The 20 µM of NPM decreased the NADH level to 12% and 4% of the level in control UACC-62 and B16F10 cells, respectively (Figure 3B). Additionally, 20 µM of NPM decreased the ATP level to 23% and 4% of the level in control UACC-62 and B16F10 cells, respectively (Figure 3C). This implies that OGC inhibition stalls the MAS and reduces NADH production by MDH2 by decreasing the amount of malate in mitochondria. To test whether the decrease in ATP was due to cell death resulting from NPM treatment, an annexin V and propidium iodide (PI) staining assay was performed with B16F10 cells treated with 10 or 20 µM NPM for 72 h (Figure 3D). There was no increase in cell death after NPM treatment for 72 h, which suggests that the ATP reduction is not due to an increase in apoptosis.

### 3.4. Antiproliferative Effect of NPM in a Human Melanoma Xenograft Model

We observed an antiproliferative effect of NPM in cancer cells, as shown in Figure 3. To test whether NPM has anticancer effects on melanoma in vivo, NPM was administered in a mouse xenograft model using the human UACC-62 cell line (Figure 4). The maximum tolerated dose of NPM was determined to be 60 mg/kg/day (PO: per os, oral treatment) (Appendix A). Three weeks of treatment with NPM reduced the UACC-62 tumor volume to about 50% of the control group (Figure 4A,C). Tumors were collected at the end of the in vivo experiment and tumors were weighed. The tumor mass in the NPM treatment group was reduced by 63% compared with the control group (Figure 4B). Immunohistochemical staining for Ki67, a marker of proliferation, was strongly reduced by NPM treatment. NPM treatment reduced the percentage of Ki67-positive cells by 67% compared with the control (Figure 4D).

### 3.5. Antimetastasis Effect of NPM in a Human Melanoma Xenograft Model

One feature of malignant melanoma is invasiveness. A previous study demonstrated that an increase in the ATP production could promote cell migration and invasion in human cancer [19]. We tested whether NPM reduces cell invasion using an in vitro Transwell invasion assay. NPM treatment significantly reduced cell invasion in a dose-dependent manner in B16F10 and UACC-62 cells (Figure 5A). We next analyzed the melanoma metastasis in vivo by inoculating C57BL/6 mice intravenously with B16F10 cells (Figure 5B). Lungs were collected at the end of the study, and hematoxylin and eosin (H&E) staining was performed (Figure 5C). The number of metastatic lung nodules and the total area of the metastases were determined using ImageJ. NPT reduced the number of metastatic lung nodules and the total area of lung metastasis by 70% (Figure 5D,E).

## 4. Discussion

In in vitro experiments with cultured melanoma tumor spheres, NPM suppressed mitochondrial activity, oxygen consumption, and ATP production in a dose-dependent manner by inhibiting OGC, resulting in decreased melanoma cell proliferation. NPM also reduced melanoma cell invasion by over 70% in a human melanoma metastasis model. Therefore, blocking OGC activity with NPM may be a useful approach for inhibiting cancer growth.

However, there are reports of the inhibitory effects of NPM on pyruvate transport, telomerase activity, and Bak protein activity. It is known that pyruvate transport may be inhibited by thiol-blocking reagents such as iodoacetate and NPM [20]. However, in cancer cells, pyruvate transport into mitochondria is limited because cancer cells have higher expression of lactate dehydrogenase (LDHA), which catalyzes pyruvate to lactate, compared with normal cells [21]. Furthermore, cancer cells do not rely on the tricarboxylic acid (TCA) cycle to produce NADH, while normal cells depend completely on the TCA cycle for NADH production [2]. Therefore, the decrease in ATP production observed in response to NPM likely does not occur through inhibition of pyruvate transport. Indeed, OCR and ATP production were measured in B16F10 cells treated with the indicated concentration of pyruvate inhibitor UK-5099 for 24 h. The analysis showed no decrease of these parameters. (Appendix A). NPM is also known to inhibit telomerase activity, which plays a key role in maintaining telomerase length [22]. In in vitro assays, NPM inhibited telomerase activity. However, the IC_50_ of NPM for inhibition of telomerase activity was approximately 2 μM, at which concentration there was no telomerase inhibition in cancer cells [22]. This implies that the cytotoxicity of NPM is unrelated to inhibition of telomerase activity. Finally, NPM has been reported to induce apoptosis through Bak oligomerization in Jurkat cells at a concentration of 0.5 μM [23]. However, we did not observe apoptosis by FACS analysis of melanoma cell lines treated with 20 μM NPM for 72 h. 

Maleimide is formed with a -C(O)NHC(O)- functional group from the reaction of maleic acid and imide. NPM is a derivative of maleimide in which the NH group of maleimide is replaced with the aryl group of phenyl. NPM can cause skin and eye irritation and showed acute toxicity, with a 50% lethal dose (LD_50_) of 78 mg/kg in mice when administered orally (https://www.cdc.gov/niosh-rtecs/ON5ACA30.html). In this study, we did not observe any death or weight loss in mice treated with 20 mg/kg NPM. Although NPM is considered a nonspecific cytotoxic agent, NPM also did not show any cardiotoxicity by hERG assay (Appendix A). The hERG channel inhibition assay is a highly sensitive assay that will identify compounds exhibiting cardiotoxicity related to hERG inhibition in vivo. The IC_50_ value of NPM in that assay was 78.75 μM, which indicates that it shows no hERG channel inhibition. 

OGC with NPM significantly reduced ATP production in melanoma cells by decreasing NADH production. Treatment of mice with NPM reduced tumor growth and tumor invasion by 60% and 80%, respectively, in a human melanoma xenograft model. These data suggest that OGC inhibition combined with cytotoxic anticancer therapy may have synergistic effects on tumor growth.

## 5. Conclusions

Biochemical data from this study using the chemical inhibitor NPM and previous studies using OGC knock-out cells [1] together suggest that proliferating cancer cells rely on the MAS system to transport cytosolic NADH into the mitochondria, where it is then used to generate ATP through OxPhos. This study suggests that OGC, as a major regulatory component of the MAS, is a promising molecular target to inhibit cancer growth and invasion by inhibition of cancer metabolism. Therefore, more efforts should be devoted to developing OGC inhibitors for cancer therapy.

## Figures and Tables

**Figure 1 pharmaceutics-12-01128-f001:**
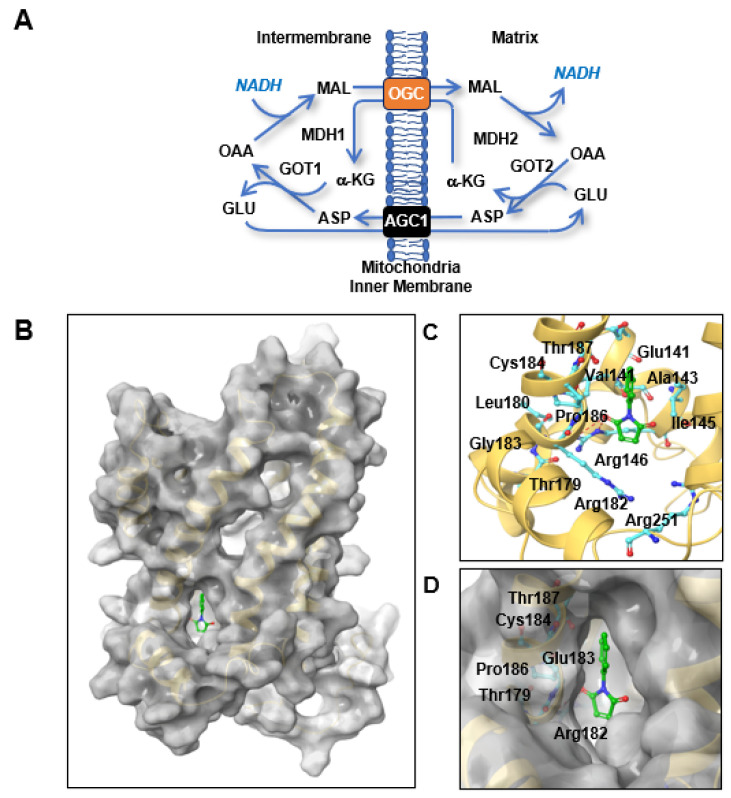
Important binding interactions for the docked conformations of NPM using the OGC homology model. (**A**) The MAS for NADH transport into the mitochondrial matrix. NPM, *N*-phenylmaleimide; OGC, oxoglutarate carrier; AGC1, aspartate-glutamate carrier isoform 1; OAA, oxaloacetate; α-KG, α-ketoglutarate. (**B**) The binding site in the OGC model is shown as a light orange ribbon; NPM is shown in green in a stick-ball structure. (**C**) Detailed interactions between NPM and OGC are shown with a stick model, and residues involved in the interaction with NPM are presented in the stick-ball style colored by atom type (C, cyan; N, blue; O, red). H-bonds are indicated by red dashed lines. (**D**) Close-up left-hand view of the predicted binding of NPM to OGC. The ligand is depicted in the stick-ball style. Figures were drawn in Maestro (Schrödinger, LLC, New York, NY, USA, 2020).

**Figure 2 pharmaceutics-12-01128-f002:**
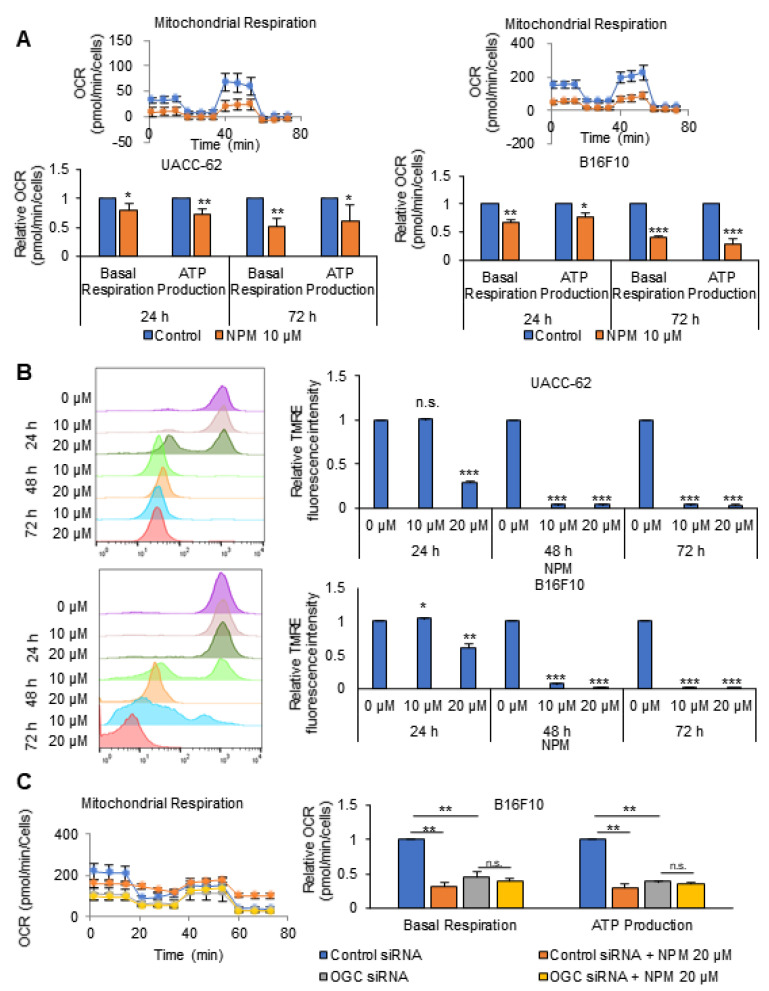
NPM reduced ATP production by decreasing the OCR and the mitochondrial membrane potential. (**A**) OCR was measured in UACC-62 and B16F10 cells treated with 10 µM of NPM for 24 h and 72 h using a Seahorse XFe96 analyzer. (**B**) The mitochondrial membrane potential was determined by tetramethylrodamine ester (TMRE) staining in UACC-62 and B16F10 cells treated with 10 or 20 µM of NPM for the indicated times. (**C**) The OCR and ATP production were analyzed by Seahorse XF analyzer after treatment of wild-type or OGC knock-down B16F10 cells with 20 µM of NPM for 24 h. Data represent the mean and standard deviation of three independent experiments. * *p* < 0.05, ** *p* < 0.01, and *** *p* < 0.001 compared with the vehicle control.

**Figure 3 pharmaceutics-12-01128-f003:**
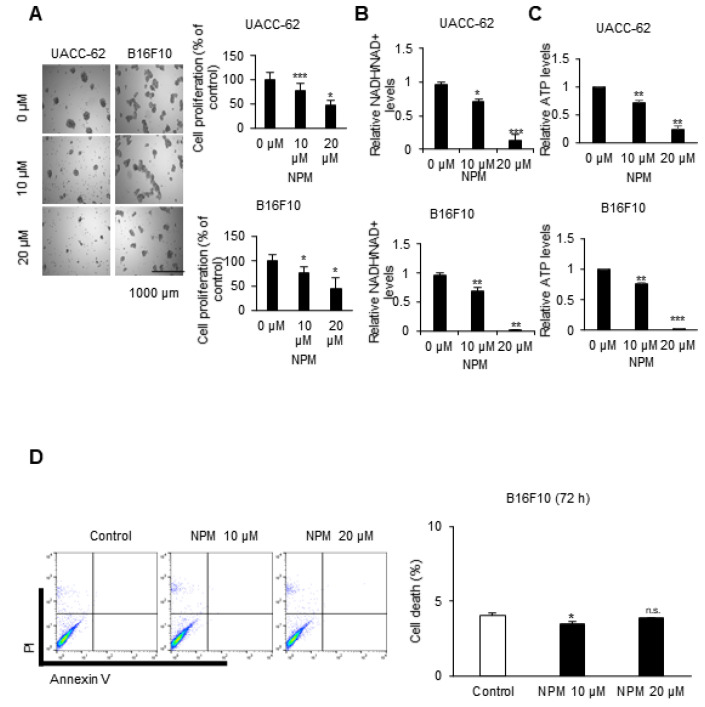
Cancer cell proliferation was regulated by NPM. (**A**) Cancer cell proliferation was determined by SRB assay in UACC-62 and B16F10 TSs treated with the indicated concentration of NPM for 72 h. (**B**) The NADH/NAD+ ratio was measured in UACC-62 and B16F10 cells treated with the indicated concentration of NPM for 72 h. (**C**) The ATP level was measured using luminescent ATP assay kit in UACC-62 and B16F10 cells following to the treatment of NPM for 72 h. (**D**) Cell death was determined by annexin V and propidium iodide (PI) staining in B16F10 cells treated with 10 and 20 µM NPM for 72 h. Data represent the mean and standard deviation of three independent experiments. * *p* < 0.05, ** *p* < 0.01, and *** *p* < 0.001 compared with the vehicle control.

**Figure 4 pharmaceutics-12-01128-f004:**
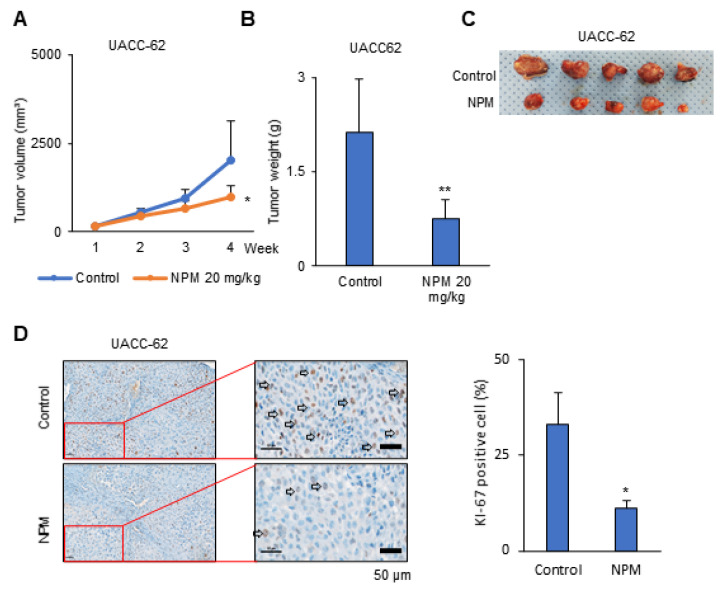
Tumor growth in a melanoma xenograft model was inhibited by NPM treatment. (**A**) The graph represents the tumor growth, as measured using calipers. (**B**) The weight of subcutaneous UACC-62 tumors from mice with or without NPM treatment. (**C**) Representative images of the removed tumors. (**D**) Immunohistochemical analysis of Ki-67 in UACC-62 xenografts from mice with or without NPM treatment. Quantification was measured by positive cell counting using Image J. Data represent the mean and standard deviation of three independent experiments. * *p* < 0.05 and ** *p* < 0.01 compared with the vehicle control. KI-67 positive cell was checked by arrow.

**Figure 5 pharmaceutics-12-01128-f005:**
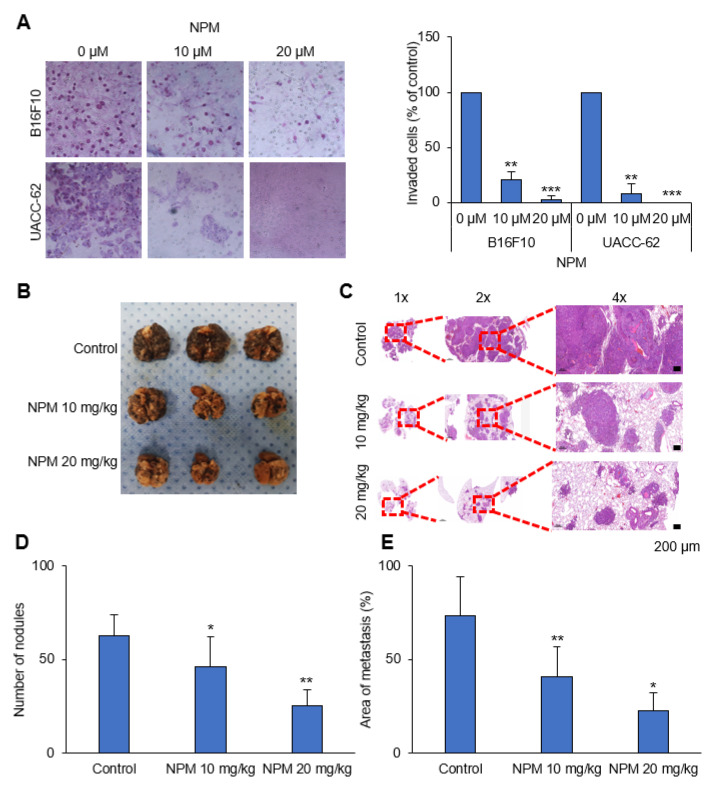
NPM reduces lung metastasis of B16F10 melanoma cells in immunocompetent mice. (**A**) Invasion assay performed with B16F10 and UACC-62 cell lines following NPM treatment for 16 h. Representative images of the invasion assay using B16F10 and UACC-62 cells (left). The numbers of invasive B16F10 and UACC-62 cells were measured using ImageJ (right). (**B**) Representative photographs of formalin-fixed lungs. (**C**) Metastatic lesions were observed with hematoxylin and eosin (H&E) staining. (**D**) Statistical analysis of the number of metastatic nodules. (**E**) Statistical analysis of the total metastatic area using ImageJ. Data represent the mean and standard deviation of three independent experiments. * *p* < 0.05, ** *p* < 0.01, and *** *p* < 0.001 compared with the vehicle control.

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
