# Peer review of "Oxoglutarate Carrier Inhibition Reduced Melanoma Growth and Invasion by Reducing ATP Production"

_pharmaceutics, 2020, doi:10.3390/pharmaceutics12111128_

Round 1

Reviewer 1 Report

In this study, the authors examined the possibility to use N-Phenilmaleimide (NPM) as an anticancer drug. The study attempted to demonstrate that NPM inhibition of the oxoglutarate carrier (OGC), as part of the Malate/Aspartate Shuttle (MAS), induced a decrease of the proliferation rate in two melanoma cell lines as well as a reduction of tumor size and metastasis invasiveness in a human melanoma xenograft model.

Overall, the work is of interest, but the authors needs to address several aspects to improve its impact and suitability for Pharmaceutics.

The issues are as follow:

Major revisions:

            Complete both paragraph 2.4 and 2.6 of the Materials & Methods section with a brief description of the protocols used. Moreover, because an optimal evaluation of the mitochondrial membrane potential is obtained using a narrow TMRE concentration range, the probe concentration must be clearly shown.

            Introduce the statistical analysis paragraph ad the end of the Materials & Methods section.

            It is not clear why the authors used the 2LCK PDB of UCP2 as template to build the 3D model of OGC. They reported in the introduction that OGC is very similar to the ADP/ATP carrier, thus why they did not start from these PDBs templates (2C3E or 1OKC)? If there is a reason, this should be stressed in the paragraph 3.1 of the Results section. On the contrary, delete the sentence concerning the ADP/ATP carrier in the introduction.

            Usually, ATP production is expressed as a rate (nmol ATP/min/ 106 cells) and the ATP level as nmol/106 cells. Here the ATP production unit is at least non canonical (relative OCR). This aspect has to be defined. Moreover, in Panel C of figure 2 the ATP production parameter is shown as relative OCR (as well as in Panel A), but in the panel C caption the parameter is mentioned as “ATP level”. Additionally, the authors reported in 3.2 paragraph of the Results section that they analyzed the ATP level (line 229). Therefore, it is not clear if the assay was done or not or if ATP production and ATP level were used as synonyms. This discrepancy must be corrected, accordingly. Finally, if the authors did not measure the cellular ATP levels, I strongly recommend adding this assay for all the conditions investigated.

            Add representative SeaHorse traces in Figure 2 to show the differences between control and treated cells.

To obtain a reliable estimation of the mitochondrial membrane potential it is necessary to measure the TMRE signal both in the absence (basal respiration) and in the presence (State 4 respiration) of an optimal oligomycin concentration. Therefore, I strongly recommend to complete Panel B of Figure 2 with the oligomycin assessment of each sample.

            Figure 3 totally lack panel C concerning the apoptosis analysis. Due to the presence of both the apoptosis protocol and the panel C caption in the text, I am expecting the results should have been in the figure. Theses must be presented.

             In general, the discussion is poor and many of the comments has to be more stressed as for example:

-the metabolic profile of the cell lines used is missing. Are they more glycolytic or more OXPHOS competent?

-the putative inhibition of the pyruvate carrier by NPM must be clearly excluded as the cause of the phenotype observed.

-another target of NPM is the Apoptosis Inducing Factor (AIF). Also discuss this aspect.

            The authors are kindly requested to explain the principle how ATP production parameter is inferred by the SeaHorse instrument. The explanation does not have to be added in the text.

Minor revisions

            At line 64 insert the term “position” before the number of the residues.

            At line 69 insert the experimental model where it has been found the reduction of 50%.

            At line 123 insert the flow cytometry model.

            In figure 4 Panel A is missing the abscissa label.

            In line 267 replace NADPH with NADH

            In line 359 change OGC with NPM or add “inhibitor” after OGC.

Author Response

November 11, 2020

Manuscript ID: pharmaceutics-988698

Prof. Dr. Yvonne Perrie

Editor-in-Chief

Pharmaceutics

Dear Prof. Dr. Yvonne Perrie

We have modified the manuscript in response to the reviewers’ comments to the best of our ability. The revised part of manuscript is marked in blue.

Soo-Youl Kim

Reviewer 2 Report

Lee and colleagues investigated the effect of N-phenylmaleimide (NPM) on cellular bioenergetics, proliferation, and aspects of tumor growth and invasiveness. They hypothesized the coordinated actions of the malate-aspartate shuttle in cancer serves to move NADH into the mitochondrial matrix for electron transport chain activity and subsequent ATP synthesis. The study presents descriptively interesting findings of potential clinical importance; however, mechanistic insight is overall lacking. Strengths of the study include the diverse collection of in vitro and in vivo cancer models to understand the therapeutic actions of NPM on in vitro cell proliferation, in vivo tumor growth using a xenograft model, and invasiveness and metastatic potential using a separate in vivo model. A major weakness of the study is the lack of experiments designed to test causality and discover mechanism. As such, the manuscript could be greatly improved by utilizing rescue experiments to solidify the general assumption that the inhibitor NPM is specific.

Major critiques:

  • In the cited study by Capobianco and colleague’s mM concentrations of NPM was required to elicit maximal OGC inhibition in a reconstituted proteoliposome system. It is unclear to me how 10 µM and 20 µM concentrations were decided for in vitro work. Extension of the observations of maleimide analogues in H460 cells may not be appropriate. If it is the case that higher levels of NPM are required for OGC inhibition, the observations may result from off-target effects.
  • If the NADH:NAD levels (Figure 3B) are being driven by decreased NADH this may confound the results of the CCK-8 assay, which reduces a tetrazolium salt to a formazan dye as a readout of cell number. If the NPM is regulating cellular NADH levels via inhibition of the malate-aspartate shuttle a better readout may be a clonogenic assay. Other assays such as MTT and Cell Titre Glo may also be acutely sensitive to NPM activity.
  • Use of NPM-insensitive OGC over expression or potentially over expression of the DIC to rescue the effects of NPM would add strength to the conclusions
  • Conclusions are overstated based on the lack of mechanism.
  • Lack of critical discussion of previously reported OGC human patient mutations (PubMed: 29431636). The findings of that study are largely is in opposition to what is presented here.

Minor critiques:

  • On line #267, I believe the authors mean to say "…NPM reduced the NADH…” rather than “…NADPH”.
  • Figure 3C is not within Figure 3 or elsewhere within the manuscript.
  • Line #288 states the maximum “tolerated” dose of NPM was 80 mg/kg/day. However, at this dosage the animals clearly die within 2 days. I would not call this “tolerated.”
  • The discussion states in line #359 that a “single dose of OGC” reduced tumor growth and invasion. However, these studies were conducted over three weeks of 5 treatments/week NPM. Moreover, on line #360 tumor invasion experiments were performed in vitro not in a xenograft

Author Response

(The authors gave the same response as above.)

Round 2

Reviewer 1 Report

The authors answered all the questions raised following the revision and considered the suggestions to improve the paper. In my opinion, after minor revisions indicated below and a supervision by a native English, the present form of the paper is suitable for publication in Pharmaceutics.

minor revisions:

  • There are parts of the document with a different line spacing. The authors are kindly suggested to uniform the line spacing.

  • Lane 64: “position at” should be “at position”

  • The sentence reported below sounds to have no meaning.

132 Mitochondrial membrane potential was assessed by measuring tetramethylrodamine ester

133 (TMRE; 87917, Sigma-Aldrich, St. Louis, MO, USA), a fluorescent probe that specifically accumulates

134 in the 100 mm plate and then treated as indicated.

I would suggest

“Mitochondrial membrane potential (MMP) was assessed by measuring the mean fluorescence intensity of tetramethylrodamine ester (TMRE) loaded cells. TMRE (87917, Sigma-Aldrich, St. Louis, MO, USA) is a fluorescence probe that specifically accumulates within mitochondria in a MMP-dependent manner. Cells were plated in a 100 mm plate and treated as indicted. Twenty minutes prior…………”

  • Lane 160: change “After end” to “At the end of…..”

  • The sentence below should be changed

361         OCR and ATP production were measured in B16F10 cells treated with indicated concentration of pyruvate

362 transport inhibitor UK-5099 for 24 h using a Seahorse XFe96 analyzer, which showed no decrease of

363 OCR and ATP production (Supplementary Figure S2).

In

“Indeed, OCR and ATP production were measured in B16F10 cells treated with the indicated concentration of pyruvate inhibitor UK-5099 for 24h and the analysis showed no decrease of these parameters.” ….or something similar…..!!

  • Lane 384: I would start the sentence with “Therefore, these data suggest……”

  • I would suggest that the different effectors added during the Sea Hoarse experiments must be indicated in the representative traces. This would clearly improve the understanding of the traces.

Author Response

Reviewer 1

minor revisions:

 The authors answered all the questions raised following the revision and considered the suggestions to improve the paper. In my opinion, after minor revisions indicated below and a supervision by a native English, the present form of the paper is suitable for publication in Pharmaceutics.

We have had English editing service.

  • There are parts of the document with a different line spacing. The authors are kindly suggested to uniform the line spacing.

- Thank you, we corrected it.

  • Lane 64: “position at” should be “at position”

- Thank you, we corrected it.

  • The sentence reported below sounds to have no meaning.

132 Mitochondrial membrane potential was assessed by measuring tetramethylrodamine ester

133 (TMRE; 87917, Sigma-Aldrich, St. Louis, MO, USA), a fluorescent probe that specifically accumulates

134 in the 100 mm plate and then treated as indicated.

 - Thank you, we corrected it as

“Mitochondrial membrane potential (MMP) was assessed by measuring the mean fluorescence intensity of tetramethylrodamine ester (TMRE) loaded cells. TMRE (87917, Sigma-Aldrich, St. Louis, MO, USA) is a fluorescence probe that specifically accumulates within mitochondria in an MMP-dependent manner. Cells were plated in a 100 mm plate and treated as indicted.”

  • Lane 160: change “After end” to “At the end of…..”

 - Thank you, we corrected it.

  • The sentence below should be changed

361         OCR and ATP production were measured in B16F10 cells treated with indicated concentration of pyruvate

362 transport inhibitor UK-5099 for 24 h using a Seahorse XFe96 analyzer, which showed no decrease of

363 OCR and ATP production (Supplementary Figure S2).

- Thank you, we corrected it.

“Indeed, OCR and ATP production were measured in B16F10 cells treated with the indicated concentration of pyruvate inhibitor UK-5099 for 24h. The analysis showed no decrease of these parameters. (Supplementary Figure S2).”

  • Lane 384: I would start the sentence with “Therefore, these data suggest……”

- Thank you, we corrected it.

  • I would suggest that the different effectors added during the Sea Hoarse experiments must be indicated in the representative traces. This would clearly improve the understanding of the traces.

- We think graphic will be busy when we add the representative traces in such as small graphic space. Seahorse method is explained in the method section as well as general method is common to every user. Therefore, we decided not to include effector traces.

Reviewer 2 Report

Lee and colleagues addressed many of the minor and methodological issues presented in the original review. 

I would like to stress the idea that correlation is not causation. It is challenging to interpret the specificity of NPM when no 'rescue' experiments were conducted. While NPM appears to correctly phenocopy OGC-knockdown according to Figure 2C and your previous publication, NPMs specificity has not been adequately addressed here.

Finally, I think it is important to discuss the previously reported OGC human patient mutations. While there may be little evidence that OGC mutation is a common genotype of all cancers, its role has been raised in the literature. For this reason, it is important to discuss your findings in the context of current knowledge of OGC in cancer.

Author Response

Reviewer 2

I would like to stress the idea that correlation is not causation. It is challenging to interpret the specificity of NPM when no 'rescue' experiments were conducted. While NPM appears to correctly phenocopy OGC-knockdown according to Figure 2C and your previous publication, NPMs specificity has not been adequately addressed here.

  • It is not fair to say that it is challenging to interpret the specificity of NPM when no 'rescue' experiments were conducted. It is a simple scientific logic; NPM inhibits OGC, OGC inhibition reduces ATP production, and NPM treatment reduces ATP production. Furthermore, we have mentioned that NPM is not specific to OGC but has several reported inhibitory effects in the paper. Although NPM itself is known to have some off-target effects, this study demonstrates that NPM showed decrease of ATP production accompanied with OGC inhibition as blocking malate transport. This implies the potential therapeutic efficacy as targeting OGC inhibition. We did not conclude NPM is a specific and good inhibitor against OGC but suggests therapeutic possibility of OGC inhibition using known OGC inhibitor NPM.

Finally, I think it is important to discuss the previously reported OGC human patient mutations. While there may be little evidence that OGC mutation is a common genotype of all cancers, its role has been raised in the literature. For this reason, it is important to discuss your findings in the context of current knowledge of OGC in cancer.

  • Reviewer continuously compels us to discuss mutation theory in this paper that we do not support any of mutation theory throughout investigation. It is a huge unsolved dilemma in oncology that I disagree with important role of mutations in metabolic enzymes. Please think about lactate production from glucose in all cancer types. LDH mutation causes this phenotype?
  • As I mentioned earlier, reviewer’s suggested reference is not a suitable to discuss about importance of OGC mutation because it is not consented with common cancer types. Paraganglioma is a rare benign tumor and metastatic paraganglioma is very rare cancer (PubMed: 29431636). Paragangliomas are rare endocrine tumors that arise from the extra-adrenal autonomic paraganglia and sympathetic paragangliomas usually secret catecholamines and are located in the sympathetic paravertebral ganglia of thorax, abdomen, and pelvis. There is no evidence that malfunction of OGC is a common phenomenon in all type of cancers. Therefore, we have right to decide not to discuss about this in this paper.
